# Lessons Learned from the Module Production for the First CMS Silicon Tracker

**Alan Honma** 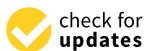

Department of Physics, Brown University, Providence, RI 02912, USA; alan.honma@cern.ch

**Abstract:** A personal view of some of the more important lessons learned from the module production for the CMS silicon tracker. This work took place from about 2002–2005. The focus is on areas where I had strong personal involvement; therefore, the tasks of hybrid production, hybrid assembly, and the wire bonding of modules and hybrids are emphasized. This article will first give a general description of the silicon tracker project and how the module production was organized. Then, there will be description of several of the key issues or problems during the production and how they were resolved. Some recommendations for future similar large-scale productions will be given.

**Keywords:** silicon strip detector modules; large scale module production; robotic assembly; front-end hybrid; wire bonding; pitch adapters; quality assurance



## 1. Introduction

The CMS experiment [1] at CERN is a multi-purpose experiment on the Large Hadron Collider (LHC), having charged particle tracking and an electromagnetic calorimeter contained inside the large bore of a powerful solenoidal superconducting magnet and with a hadronic calorimeter and muon tracking on the outside of the magnet. The tracking system consists of a silicon pixel inner detector and a silicon strip outer tracking detector (or SST for "silicon strip tracker") [1–3]. See Figures 1 and 2 for the overview of the CMS detector, location of the SST, and overview of the SST. The SST was to be the largest detector of its kind (by more than an order of magnitude) in terms of the surface area of silicon strip sensors (>200 m$^2$) compared to its predecessors. Its modular design meant the device was built with 15,148 individual modules having 29 module types. In 2000, robotic assembly of modules was demonstrated with prototypes which, along with the successful industrial production of sensors on 6" wafers (allowing for larger sensors), motivated the large increase in the volume of silicon strip modules, allowing the outer tracking detector (60 cm < R < 120 cm) to be made using silicon. Originally, only the inner tracking detector (R < 60 cm) was planned in silicon. The robotic assembly was based on a large high-precision computer-controlled industrial gantry device, custom outfitted with tooling adapted for module assembly, as seen in Figure 3 [4]. It was also decided to de-centralize module production and wire bonding (needed to make the micro-connections between the sensors and the readout electronics), with 6 robotic (and one manual) module assembly centers and 14 wire bonding centers that were created in various collaboration institutes. At CERN, only the assembly of the readout components of a module would be produced, using the original gantry that was used to demonstrate the robotic module assembly principle. The industrial scale of component production (performed primarily in industry) and module production (performed at the collaboration's assembly centers), the bulk of which took place between 2002 and 2005, made the project very complex in terms of management, logistics, and quality control. The project was pioneering in many aspects and some of the critical lessons learned that occurred during the module production are the subject of this paper.

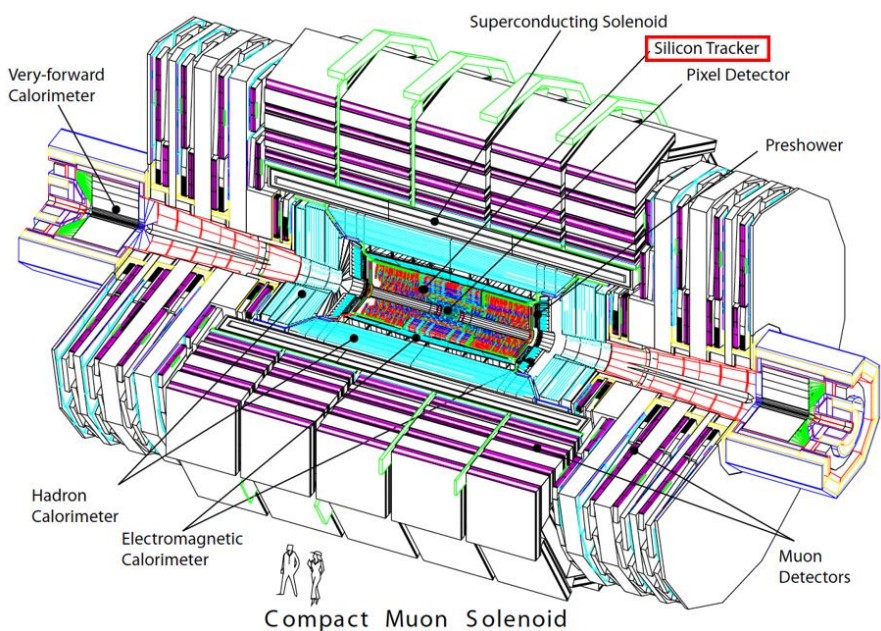

**Figure 1.** Cut-away view of the CMS detector, the SST is indicated as "Silicon Tracker".

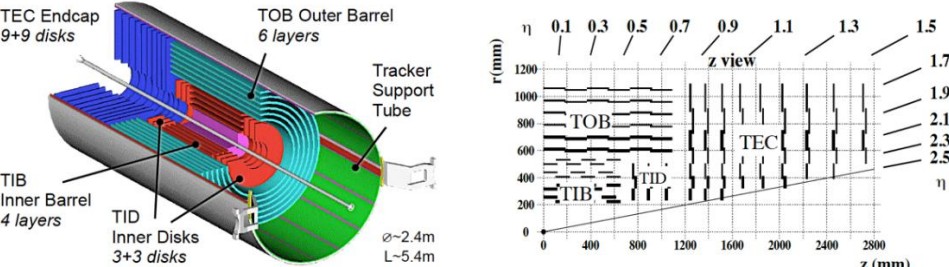

**Figure 2. Left**: Cut-away view of the SST on left. **Right**: A cross-section of a quadrant of the SST showing the position of the modules as the line segments.

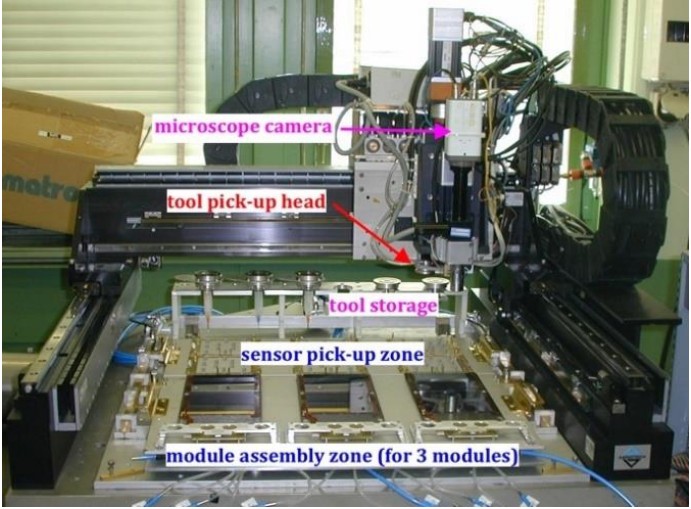

**Figure 3.** Prototype gantry system demonstrating module assembly of three modules.

## 2. Some Technical Details of the CMS Silicon Strip Tracker

In order to better understand some of the key lessons learned during the module production, a few more technical details about the tracker construction are given below.

The CMS silicon strip tracking detector was based on single and double sensor modules. In the case of double sensor modules, the two sensors were positioned next to each other such that the corresponding strips (there were either 512 or 768 strips on each sensor) could be electrically linked to each other and then connected to the readout electronics at the end of one of the sensors (Figure 4). A single sensor module was simply connected to the readout electronics at one end (Figure 5). The front-end readout electronics consisted of a flex circuit mounted on a ceramic substrate, known as a front-end hybrid or just "hybrid". All the inter-sensor and sensor-to-electronics connections were performed by means of aluminium wedge wire bonding, which will be described in more detail later. The other key components of a module were the light weight and thermally conductive frame (made of either carbon-fiber laminate or of graphite) and a fairly simple high-voltage biasing flex circuit, which was glued to the frame. When assembling a module, the robot needed to dispense glue in the appropriate places and then to pick and place the components (sensors and hybrids) from a staging area onto the frames. The needed accuracy of placement was of the order of 5 μm, which was challenging given the large working area of the robot (about 50 cm by 50 cm).

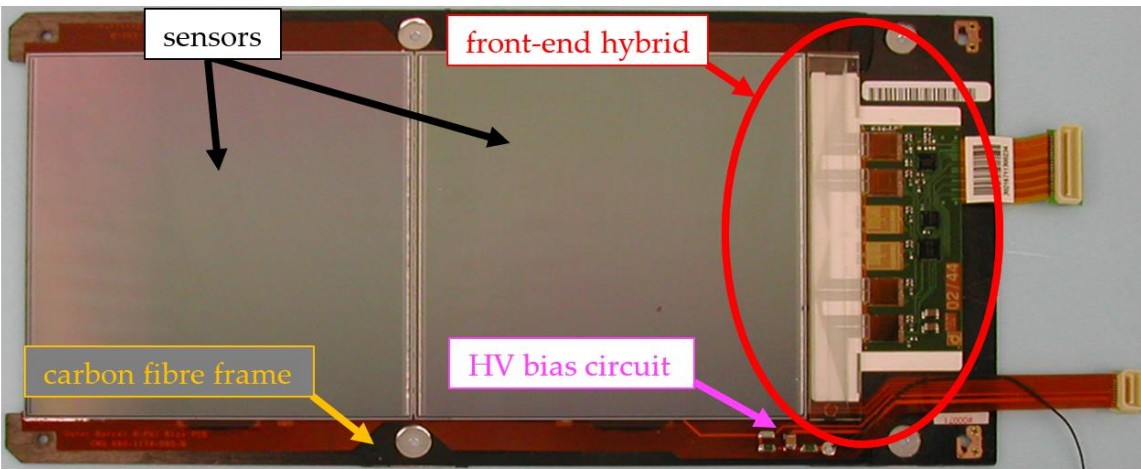

**Figure 4.** Photo of a two-sensor module showing its basic components.

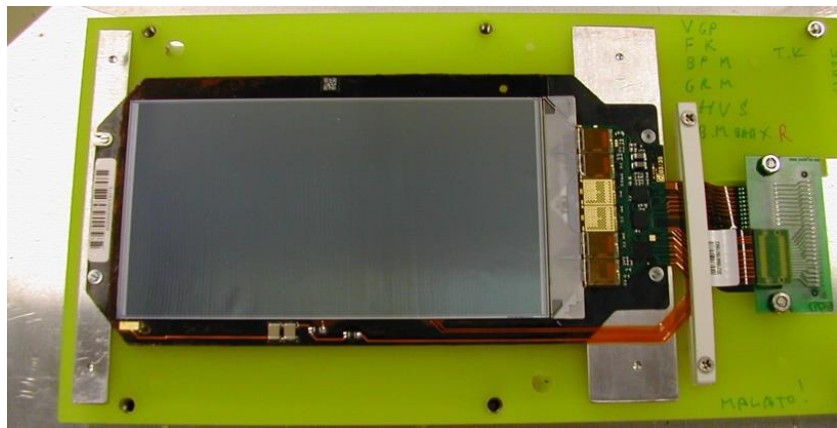

**Figure 5.** Photo of a single-sensor module.

After the curing of the glue, the micro-connections between the sensors (in the case of two sensor modules) and between the sensors and the hybrids were performed using ultrasonic aluminium wedge wire bonding machines. These machines are regularly used in industry, primarily to connect bare electronic chips made on silicon to their carrier printed circuit boards (PCBs). After wire bonding, these chips are then encapsulated in a ceramic or plastic package, which are the familiar "chips" found soldered to PCBs seen

in all electronic devices such as watches, phones, televisions, computers, etc. In the case of the modules, these wire bonds are usually left exposed because of the very large areas they occupy and the fact that a detector module is meant to be sealed in a well-controlled environment that is not subject to the handling and mechanical rigors experienced by many industrial PCBs. The need for between 500 and 1500 wire bonds per module, with more than 15,000 modules needed, meant that around $15 \times 10^6$ wires were needed for these connections on the modules. Add to this the fact that wire bonds were also needed for making the hybrids (see below), a total of more than $25 \times 10^6$ wires were required for the total project. This was another of the major challenges since this amount of wire bonding was at least an order of magnitude more than in any previous silicon detector. Moreover, wire bonding, although a standard process in the micro-electronics industry, was new at this very large scale for high-energy physics institutes.

On the hybrids (see Figure 6), the heart of the readout electronics consisted of ASIC (application-specific integrated circuit) chips known as APV25 chips, which contained 128 channels of charge-sensitive amplifiers followed by a multiplexing circuit that serialized (in time) the 128 analogue output signals. There were either four or six APV25 chips mounted on each hybrid. Because the pitch of the input bond pads on the APV25 chips did not match the various different pitches of the sensors, a "pitch adapter" was needed. This was an aluminium on glass circuit, which had the same bond pad pitch as the APVs on one side and the same pitch as the sensor on the other. The pitch adapter was glued next to the APVs on the hybrid's ceramic support. The APV25 chips were glued to the flex circuit which contained a few other small packaged chips and a large number of miniature surface mount components (SMDs). One end of the APV25 chips was wire-bonded to the flex circuit to provide the power, control, data output, and monitoring signals. The other end of the APV25 was wire-bonded to the pitch adapter. A completed hybrid consisted of this flex circuit with its APV25 chips and the pitch adapter, which were both glued to an alumina substrate. All of the flex circuits on their ceramic substrates but with no pitch adapter (a "bare hybrid") were made in industry and the wire bonding between the APV25 chips and the flex circuit was performed by the assembly company. This allowed only "tested good" bare hybrids to be delivered to CMS. All of these circuits were delivered to CERN, where they were visually inspected and re-tested electrically, and then the CERN gantry was used to pick and place the appropriate pitch adapter and glue it on the ceramic substrate next to the APV25 chips. Then, half of these hybrids were sent to a collaborating institute and half were kept at CERN for the wire bonding between the pitch adapters and the APV25, chips as shown in Figure 7. When the wire bonding and an electrical re-test of the hybrids was completed, they were shipped to the module assembly centers.

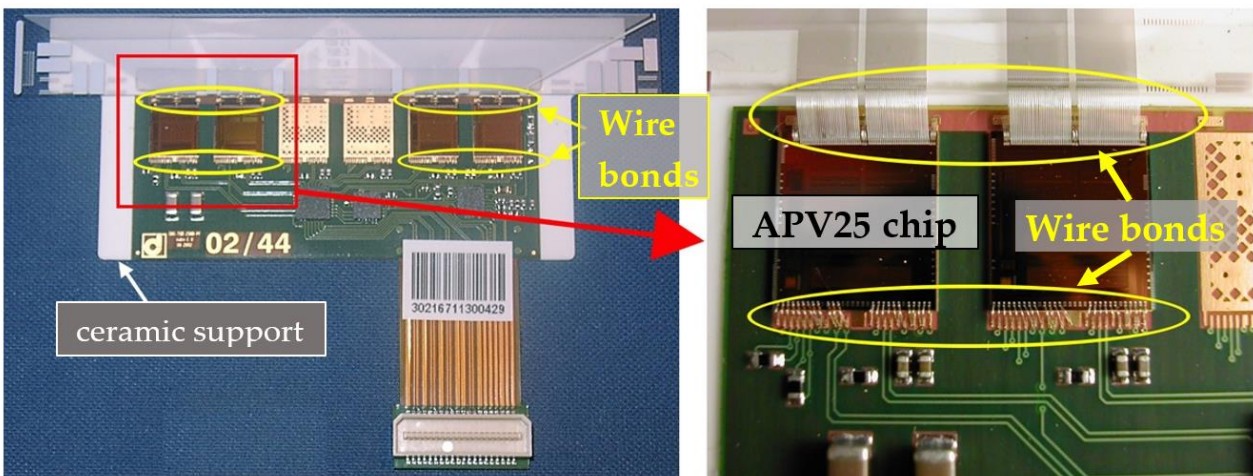

**Figure 6. Left**: Hybrid (flex PCB+pitch adapter on ceramic substrate). **Right**: Zoom on two APV chips showing wire bond connections of chip-to-flex (lower row) and chip-to-pitch adapter (upper row).

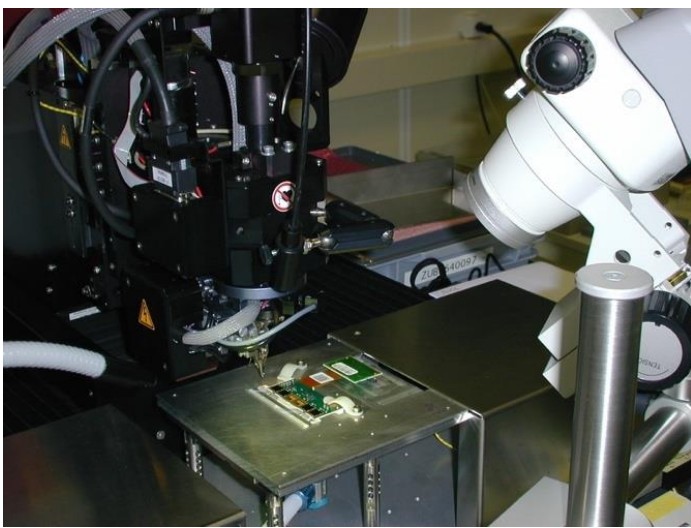

**Figure 7.** Wire bonding machine in the process of making chip-to-pitch adapter bond wires.

### 3. Lessons Learned

Out of the many issues resulting in useful lessons learned during the module production, six main lessons will be presented below. They are a personal choice of example lessons based on importance and my direct involvement (hence a more direct knowledge of the issues).

#### 3.1. Lesson 1: Robotization of Module Production

This first lesson, unlike the others, is not based on a problem that occurred during production that required a solution. It is rather a look at what was learned in this first attempt at high-quantity robotic module production in this field [5]. Despite the fact that the module production was spread amongst six centers using eight gantry robots in total (and one center performing manual assembly), this still implied an assembly rate of approximately six modules per day given the initial production time of 2 years. In fact, the allotted time was reduced to something closer to 1.5 years owing to delays in component availability, so more typically 10 modules per day were needed. The working space under the gantry determined that only three or four modules could be built at one time, so a center with one gantry would have to build three or four sets of modules per day. The task was performed on time but, during the prototyping and start-up period, it was found that a lot of changes and re-organization would be needed. Since the CERN gantry was also performing robotic assembly, but of pitch adapters onto the hybrids, some of the lessons learned came from this experience as well.

The first point in the robotization "lesson" is that the design of the module needed to be very well adapted to robotic assembly. Much thought went into this because previous module assemblies were performed by "hand" and a person can more easily adapt to the unexpected. For instance, the positioning of the components prior to assembly had to take into account that the gantry camera could find the precision markers on the components using pattern recognition in order to calculate the corrections (in x, y, and rotation) needed to place the component with high accuracy. For manual assembly, the technician would normally simply pick and place the component by hand and then use high-precision adjustment stages to get the final accurate position. Thus, the design of the components had to take into account suitable markers for pattern recognition and have reasonably precise locating holders for the components. A safe method of pick and place performed by a robot was also needed (this was accomplished with a number of custom pick-up tools using vacuum). Another key point was that it was necessary to ensure that the parts placed with still-uncured glue did not move. This was also accomplished with a vacuum system but this point was particularly difficult. In order to get a good vacuum on the components,

the component needed to be very flat and smooth. This was usually but not always the case. However, a failed vacuum on one component could jeopardize the vacuum on other components since individual vacuum lines for each component were not feasible given the large number of components. It was found that a flexible suction cup system worked the best for achieving a good vacuum even with components with poor flatness.

It was also realized that the activities surrounding the robotic hybrid and module assembly process often required much more time and manpower than the assembly job itself. These activities included: defining the required type of module for assembly; finding, inspecting, and placing the components on the gantry assembly tray; preparing the glue(s); initializing the gantry and setting up the appropriate program based on the module type under assembly; inspecting (visually) the result of the assembly; moving the completed tray to the glue curing area; and cleaning up the glue preparation tools and the fixtures. Should anything go wrong during the assembly, the operator may also have to stop the assembly, fix the problem (such as failed vacuum on a component), and then restart the program at the appropriate place. Many more tasks were needed outside of the above-described activities in order to keep the production line going and some of these will be discussed in Section 3.6.

In the end, the module and hybrid assembly production were accomplished within the time period required and the resulting quality of the modules was quite good, despite the many small hiccups and despite the more serious problems in obtaining good-quality hybrid circuits from the industrial suppliers, to be discussed in Sections 3.2 and 3.3.

### 3.2. Lesson 2: "Via" Problems in the Flex Circuits

This second lesson concerns a problem that occurred during the production and resulted in a considerable delay in the bare hybrid delivery, which therefore delayed the module production. A bit of additional technical information is needed to best understand this problem and the solution.

The bare hybrid, as delivered to the collaboration [6], was manufactured by two different companies, one of which made the flex circuit (PCB) and then laminated it to the ceramic substrate. The second company (the assembler) placed and soldered the surface mount components; glued the APV chips; and then wire-bonded the control, power, and readout lines from the chips to the PCB bonding pads. The assembler also performed an overnight passive thermal cycling of the completed hybrids followed by a quick electrical test using a tester provided by CMS. The flex PCB was a very complex one for the time and pushed the flex producer close to their technological limit. In particular, the circuit was a four-metal layer circuit, containing both buried and through "vias". Vias are plated through holes between layers used to pass signals or power from one layer to another. What was particularly difficult in our circuit was that the metal line width and spacing were very small and the via size was also very small (the hole drilled for the vias was 100 μm in diameter). The flex manufacturer assured us that they had successfully made other PCBs with similar via structures previously and thus had the technological capability. Therefore, we went ahead with this production with confidence, after a successful prototyping phase.

As part of the standard quality assurance plan for PCB products, the flex manufacturer was required to conduct complete flying probe electrical testing of each bare PCB before it was allowed to be used in the assembly step. At this point, it could be seen if breaks in the PCB lines or failures of vias had occurred. Any non-continuity of lines or short-circuits of a line to another would be detected and the PCB rejected. We monitored the rate of such failures (the bare PCB yield) because a poor yield could indicate a systematic problem such that more or all PCBs may fail with time. No such yield problem was detected with the prototype runs of the PCB and only a small rate of failure was observed at the start of the main production. However, when the first completed hybrids were received by CMS and further testing (including after thermal cycling) was performed on hybrids, it was found that some chips or complete hybrids failed, although they had passed the initial electrical tests at the company and on reception at CERN. Although the rate of failure was

still small (several hybrids out of about one hundred tested), given that the hybrids and modules would experience much more handling and power cycling and thermal cycling before going into the final detector, and should be able to work for 10–15 years in both cold and ambient temperatures, it was clear that even this small rate of failure was a potential disaster and could not be tolerated. A failure rate of much less than 1% was required. Therefore, the hybrid production was stopped and a crash program to understand and fix the problem was undertaken.

From the evidence of how the failure was seen in the electrical tests, the problem was found to be an open circuit that occurred in one key line in the hybrid. A simple electrical continuity test found this open circuit. The only likely reason for the open circuit was a failure in a via on that line, which had to connect one metal layer to another. Only a few lines on the hybrid that passed through vias were as critical as this one, without which one of the APV25 chips would not function correctly. Several of the hybrids with this problem were then sacrificed by being cross-sectioned to reveal the via's structure. What was found was quite surprising, there was very little metal in parts of the via. The metallization of the via hole should produce a cylinder of copper on the circuit's surface where the hole was made. This cylinder should be around 5 μm thick (or more) at all points on the cylinder surface. Cross-sections were made through the vias in the PCBs in many locations such as the one indicated in the right photo of Figure 8. As can be seen in the middle photo in Figure 8, the cross-section showed that the metal skin was not at all uniform and in many places the thickness was much less than 5 μm and in some places there was no copper at all. It was clear that such vias were not correct and could easily fail with time and especially with thermal cycling, which would stress the metal cylinder and cause already thin zones to crack and fail. Since in most of these poorly metallized vias there was still a thin amount of metal such that it would still conduct properly between the layers, the bare PCB would pass the open/short electrical tests. After assembly (which meant some mechanical stress and a large thermal one from the solder reflow process), a small number of electrical failures occurred on completed hybrids but up until then these were thought to be due to more random faults such as ASIC chip failures, line breaks, and poor soldering. A more careful examination revealed that many of the electrical failures were due to these vias going open circuit.

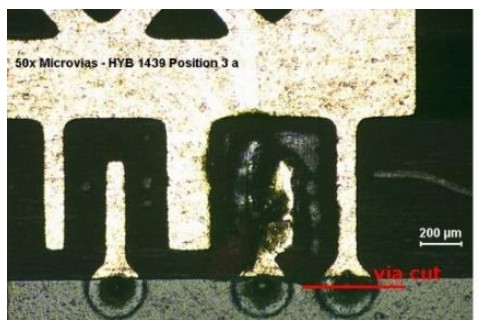 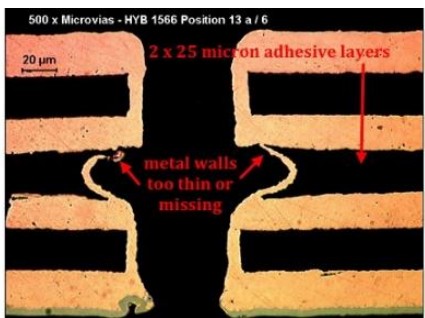 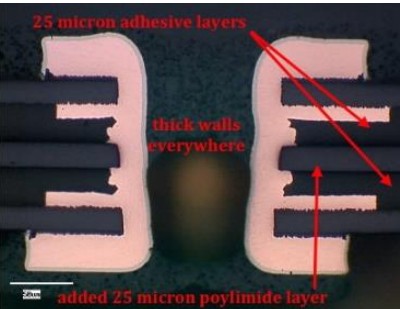

**Figure 8. Left**: Location of via cross-sectional cut. **Middle**: Original poorly plated via walls. **Right**: Improved solution giving correctly plated walls.

At this point, there were a number of critical issues to resolve:

1. Why was the via metal cylinder wall not straight and not plated correctly?
2. How to obtain a properly plated via?
3. How to assure that much fewer than 1 in 1000 vias would have a failure?
4. How to assure that the vias would survive the thermal and mechanical stress to be expected in the 10–15 years of the detector's life?
5. How to assure that no other similar problem would cause hybrids to fail?

The discovery that the problem was a poorly plated via problem took several weeks. The attempt to understand how the problem occurred took several months but, in the

meanwhile, the PCB manufacturer tried a number of modifications in order to try to solve the problem even though they did not understand the actual cause. As it turned out, the studies to understand how the problem occurred concluded that the drilling of the vias caused an uneven hole wall to be made such that the adhesive layers in the stack-up were more abraded than the polyimide layers, giving the concave-like metal shape along the adhesive layers seen in the cross-sections. The solution that was chosen was to add a polyimide layer between the two adjacent adhesive layers in the center of the stack, thus reducing the adhesive thickness from 50 to 25 μm (all other adhesive layers were only 25 μm). It was observed that the very poor metal thickness in the cross-sections occurred only at this thicker adhesive layer. In addition, the metallization process was modified to deposit more copper into the vias. The combination of these changes resulted in vias with much better metal thickness and uniformity, as can be seen in the right photo in Figure 8.

The above modification addressed only points 1 and 2 above. To address points 3 and 4, it was decided to add a "test coupon" to each PCB panel, which contained via "daisy chains". The via daisy chain consisted of a very large number (several hundred) of all the types of vias used in the real circuit where a signal line passed in series through all of these vias such that one could see clearly if any of the several hundred vias went open-circuit using a single continuity test. Several of these via daisy chain structures were placed in different locations on the PCB panel. The test coupons could be removed from the panel and thus were subjected to several passes through a solder reflow cycle, giving them even more stress than the circuit would see during the soldering of the SMD components. In addition, we required that these test coupons see cold thermal cycles as well since the operational temperature of the CMS tracker was expected to be about −20 °C at the later part of the detector lifetime (needed to minimize the radiation damage to the silicon sensors). Then, each coupon would have the via daisy chain tested for continuity and also a resistance measurement performed since this could reveal a weakening or reduction in the amount of metal in the vias even though it was still conductive. Finally, we required that all assembled circuits be subjected to five passive thermal cycles of −20 °C to +30 °C and an electrical test be performed after those thermal cycles.

With the above-mentioned modifications, the via failure problem was effectively "solved", and the resulting batches of the PCBs showed very few failures in the via daisy chain test (although there were a few batches of bare flex circuits that were rejected because they failed this test). Moreover, very few failures were found after assembly and thermal cycling but those hybrids that failed were not accepted and a low batch yield would reject the batch. It is true that some of the early batches of hybrids were nevertheless accepted and used in the final detector because we conducted extensive thermal cycling and electrical testing and found the failure rate to be low (and also because we were so far behind schedule that we desperately needed some modules for integration and long-term module tests). However, some 4000 hybrids (out of a total of 16,000 needed), mostly in batches made prior to the fix, were rejected because of known or suspected via weakness and this problem delayed the project by about 9 months.

There is an ironic post-script to this issue: the via failure cause was, in fact, not diagnosed correctly, although the solution was effective, nevertheless. I worked on another CMS silicon detector project just following the tracker construction, the CMS pre-shower detector. It involved the building of about 4400 modules of what one could call silicon pad detectors (or a silicon strip detector with very wide strips, 2 mm compared to 100 μm). However, it also used a complex multi-layered hybrid for its front-end electronics that used a combination of flex and rigid layers in its build-up. It also had many via structures similar but slightly larger in size to the tracker hybrids. It also had some thick adhesive layers and it was found that some of the vias were failing in a similar way to the tracker case. Cross-sections showed nearly the exact same concave cavities in the adhesive layers. The flex manufacturer (a different one than for the tracker) conducted an even more extensive study and learned that these cavities were not due to the drilling of the vias but rather the plasma etching step used to "clean" and prepare the vias prior to the copper plating step.

The plasma etching ate away the adhesive much faster than the polyimide, thus creating the concave cavities in those layers. The solution was quite similar to that of the tracker, minimizing the thickness of the adhesives and increasing the thickness of the copper plating. However, since they knew that the problem was from the plasma etching, they could also try modified etching parameters that would reduce the size of the cavities while preserving the quality of the cleaning and hole preparation. It was satisfying, however, to finally understand the true cause of the via failure in both projects.

There were multiple lessons from the via failure problem. One was not to assume a company had mastered a difficult technical feature simply because they claimed to have performed something similar (but not identical) to what you require. Second, for vias and other delicate PCB structures, test coupons with via daisy chains and copies of the other delicate structures (such as very fine lines or small bond pads) should be included on the PCB panels. They should be electrically tested (if possible) and, in many cases, cross-sectioned if there is a risk of incorrect metal plating or non-conforming shape or thicknesses.

### 3.3. Lesson 3: Wire Bonding Problems in the Flex Circuits and in the Modules

As the amount of wire bonding needed in this project was massive and occurred in several steps in building the module (first at the hybrid assembly company, second at two CMS centers for bonding the pitch adapter to the readout chips, and finally at the 14 CMS bonding centers to bond the sensor-to-sensor and sensor-to-pitch adapter; see Figure 9), there were many problems, most of them minor, that needed to be resolved. However, there were three major problems that led to important lessons.

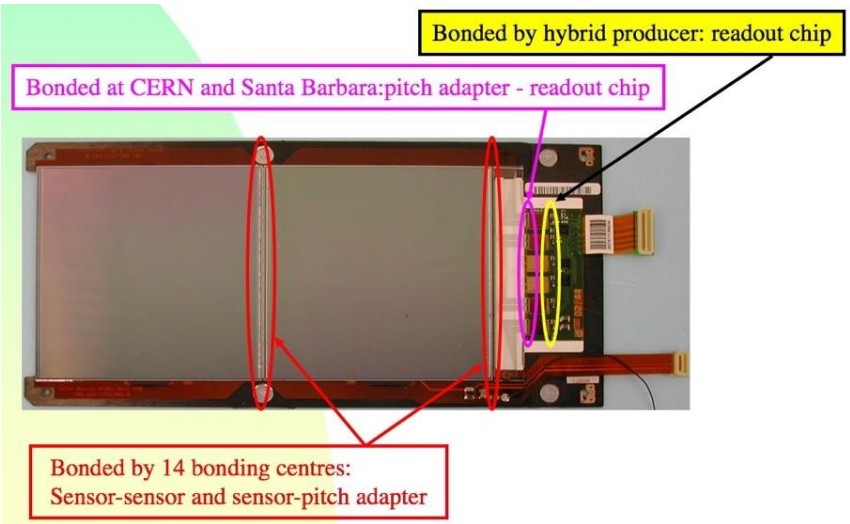

**Figure 9.** Location of wire bonds in a two-sensor module.

### 3.3.1. Wire Bonding Problems #1: Weakened Wires

The first problem occurred in the front-end hybrids, where the assembly company performed the wire bonding from the readout ASICs to the PCB bond pads (see Figure 10). We had (fortunately) decided to ask the company to make "dummy" hybrids (electrically bad but mechanically good) with dummy chips so the soldering, gluing, and bonding steps could be tested without wasting good parts. One out of twenty hybrids assembled was a dummy. Both the company and CMS conducted wire-bond pull tests on the bond wires they placed on the dummies. A pull test is when you take a dynamometer with a very tiny wire hook and pull up on a bond wire to measure how much force it can take before it either breaks or the wires detach from one of the end weld points (called the bond foot). The problem that occurred was that CMS started to get very poor results on the pull tests of some of the hybrids that the company had also tested but with good results. The difference

was very substantial in some cases: the company got nominal results (which was a pull test strength of about 10 g) whereas CMS would get values of 2–3 g (the required value was a mean of 8 g for a set of 10 wires).

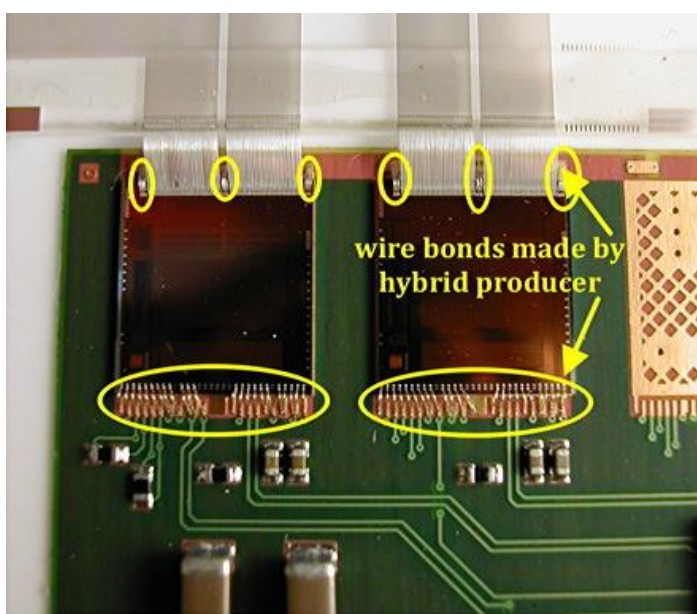

**Figure 10.** Location of wire bonds on a hybrid assembly made by the hybrid producer.

After several heated discussions about whether the company or CMS was conducting the test correctly, it was found that the result depended very strongly on precisely where the wire hook was placed when conducting the test. The company had chosen to pull very near the apex of the wire loop whereas CMS was pulling closer to the central point mid-way between the ends of the wire (a difference in distance of perhaps 0.3 mm out of the typical 3 mm length of a wire). However, this very small difference in pull location could lead to very large differences in the result. There were good reasons for choosing either position; the company's choice would result in the least change in position of the wire during the pull test (moving the wire will generally weaken it) whereas the CMS choice would result in the force applied to the two ends of the wire to be nearly equal, thus testing both ends fairly.

There were more heated discussions as to which was the correct method to use in order to apply the pull test criteria and decide if the wire bonding passed or failed those criteria. As CMS was the client and because we wanted the wires to be able to survive the handling and thermal changes over the long lifetime of the detector, we insisted that if a wire is slightly moved during this test, it should not lose 90% of its strength. In fact, a more careful analysis of why this weakness occurred after such a small flexing of the wire showed that because of the design of the ASIC, the wire was developing a crack at the point where the wire rose upward from the ASIC bond pad. This crack developed because there was a "passivation" (protection) layer of polyimide, which was about 7 μm thick and surrounded the ASICs' wire bond pads. Because the bond pads were very small (95 μm × 95 μm), the bond weld foot (which was about 75 μm long) often would run into the edge of the polyimide, which could create a dent in the soft aluminium wire and the local stresses and further movements of the wire during the bonding process caused a crack to form near this point (see Figure 11). If the wire was not much stressed further (as for the company's method of pull testing), the strength of the wire was close to normal. If the wire was moved physically (as was the case for the CMS method of pull testing), the crack would quickly propagate through much of the rest of the intact wire material and cause a failure with very little (or even no) pull force.

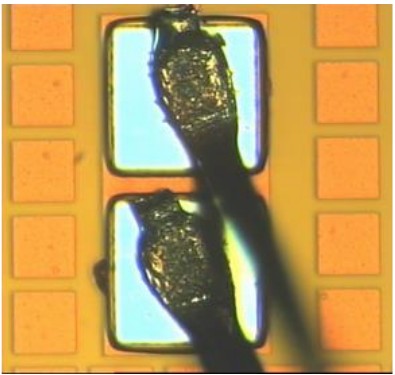 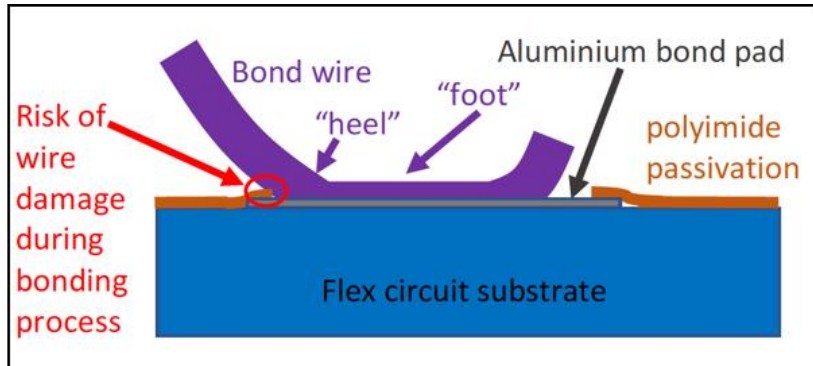

**Figure 11. Left**: Photo of wire bonds on an APV25 chip made by the hybrid assembler. **Right**: Drawing of a side view of one wire bond on an APV25 chip indicating the location of the probable cause of weakening of the wire.

The solution was to use a wire bonding tip that reduced the length of the bond weld foot and to position the placement of the foot to avoid hitting the edge of the polyimide. These solutions did work but prior to that one large batch of about 500 hybrids was rejected because of weakened bond wires. The problem could have been easily avoided if the bond pads had been made a bit larger and the polyimide passivation window had also been made larger. Unfortunately, the chip design was created years before the production and could not be changed. In prototype testing of wire bonding to the chip, this was performed by CMS and the problem of hitting the edge of the polyimide had not occurred, so the potential danger was not identified. It is true that the bond pad size was marginal and could have been made slightly larger without a major re-design of the chip. We note that future chip designs should allow for more space for the bond foot or a thinner passivation layer should be used (which is now the case, 1–2 µm is more typical of recent ASICs). It was also important to realize that if there is a chance of wires being moved, a pull test that minimizes wire movement may not reveal crack damage that could lead to severe weakening and potential failure in the future.

3.3.2. Wire Bonding Problems #2: Shipping Damage to Bond Wires

In an early two-sensor prototype module that was built in the United States and transported by air to CERN by a CMS collaborator, it was found to have nearly all its bond wires connecting the two sensors and connecting the sensor to the pitch adapter broken or severely weakened. The module was carefully handled during the whole transport process (it was in the person's hand baggage) and it was certain that it was never subjected to severe shocks such as what one might expect if the baggage was dropped or hit hard by another object. The hint that this damage might have been caused by vibration came from the fact that the broken wires were the ones in the center of the row of wires and not near the edges, where the sensors were glued to the module frame. The sensors were only glued to the support frame at the four corners and thus the central parts of the sensor edges were unsupported and could thus be moved more easily. However, the sensors were 500 µm-thick silicon, which is quite rigid, and thus it did not seem like they could bend sufficiently to cause movement damage to bond wires.

In order to study the possible cause of the damage, test modules were taken to a lab at one of the collaborating institutes where they had vibration test equipment. It was found that there were a number of resonant modes of a sensor held in the way it was held in the CMS modules but the most important one was the primary mode where the center of the sensors moved up and down. Since the fixed points were the corners, the unfixed edges would also move up and down at that resonant frequency although with a smaller amplitude than the center of the sensor. However, the amplitude of movement at the midpoint of the edges could reach millimeters, even with a very small driving amplitude of the vibration system. The frequency of resonance was around 500 Hz. In studying the

typical frequencies found in transport (motor vehicles, trains, and airplanes), previous research showed that in the range from 1 Hz to about 2000 Hz transport vibrations could be significant. The higher frequencies (above 200 Hz) were more typical for air transport and the lower ones for ground transport. Thus, it was concluded that, in the airplane flight from the US to Europe, there must have been some period of time where the module experienced vibration frequencies very close to the main resonant frequency. The sensors would then have moved with large amplitudes and because each sensor would have slightly different resonant frequency (and because the pitch adapter probably did not move much at all since it was well-glued to the module frame), these parts with wire bonds attached were not moving in unison and thus could have experienced many bending movements, which would primarily weaken the point of attachment of the wire to the substrate, which is normally the weakest point of the wire system (owing to the movement experienced by the wire during the bonding process itself).

Since the modules would be bonded in 14 different sites in both the US and Europe, they would be seeing quite a lot of transport before being finally installed in the CMS pit in the LHC. Therefore, a solution was needed to avoid any such transport damage. One solution to this sort of damage has been used in industry for nearly as long as wire bonding has been used for chip attachment to its package. This solution is called potting or encapsulation and consists of encasing the chip and its wire bonds in a material that is applied as a fluid and then polymerizes or cools to become a solid (see Figure 12). Quite often the material used is a type of epoxy adhesive which hardens to something very tough and robust. However, this was not a good solution for the CMS modules as there was evidence that large surface coverage of epoxy adhesives on the active face of silicon sensors could damage or change the behavior of the charge collection of the sensor, thus degrading its use as a particle detector. It would not have been such a problem had the detector been intended for room temperature use only. However, the CMS silicon modules would need to operate at −20 °C and, because of the large coefficient of thermal expansion (CTE) of these adhesives (typically 100 ppm/K compared to 3 ppm/K for silicon), there would be a very large stress on the silicon surface caused by the adhesive shrinkage when cold. Thus, it was decided to use a flexible silicone-based adhesive instead. This still had a large CTE but, because it remained flexible even at low temperatures, it would not create a large stress on the silicon surface. Many tests were performed to check that this material did protect the wires from vibration damage and yet did not overly stress the sensor at ambient or when cold. It also had to be tested for radiation hardness since it would receive a significant amount of radiation during the lifetime of the silicon tracker in LHC. In addition to this encapsulation, the sensor-to-sensor and sensor-to-pitch adapter edges were glued to a bridging piece of ceramic (which had a low CTE) in order to prevent these parts from moving with respect to each other. It was decided to apply the flexible encapsulant only for the modules made in the US, since those were the ones to experience air transport. The modules made in Europe had the extra support but not the encapsulation. There was no known transport damage to bond wires in any of the modules made during the production phase, so these solutions did work well.

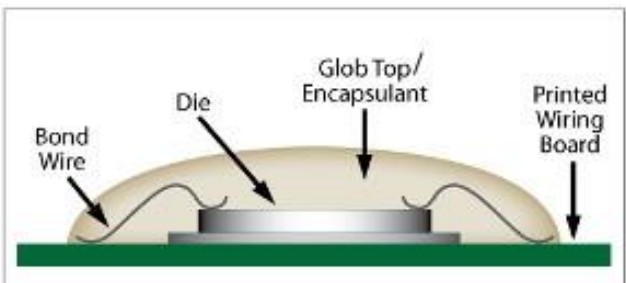 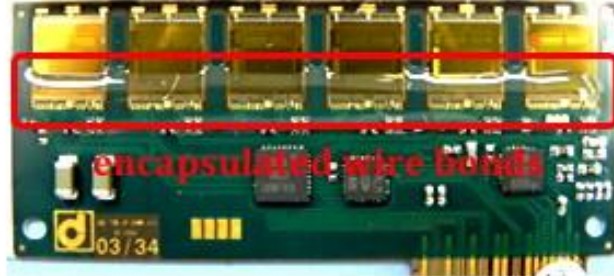

**Figure 12. Left**: Principle of encapsulation of wire bonds. **Right**: Photo of encapsulant on hybrid.

### 3.3.3. Wire Bonding Problems #3: Handling Damage to Bond Wires

As explained in Section 3.3.2, the modules made in the US (about half of the total) had their bond wires protected by encapsulation. In fact, the encapsulation was also applied to the wires made by the hybrid assembler for the US-made modules, just for protection from handling (an accidental touch by personnel or even from a light touch from packing material could easily break wires). However, no encapsulation was applied anywhere on the modules made in Europe. The CMS tracker was composed of three main sections, a "barrel" middle section and two "endcaps" mounted at each end of the barrel section. A large fraction of the modules was mounted for one endcap at CERN. It turned out that the mounting of the endcap modules was especially tricky as a lot of hand manipulation was needed to place and screw down the module. This manipulation required the operator to have their fingers and tools very close to the sensors and hybrids of the modules.

During the endcap module mounting work at CERN, a very large number of modules (about 400 in total) had their bond wires either damaged or broken. Several modules had the sensor broken and were thus non-repairable. However, those with only the bond wires damaged were usually repairable and could be re-installed in the endcap structures. The bond wire damage (an unrepairable example is shown in Figure 13) was mostly to the wires going to the ASIC chips, either the signals coming from the sensors (via the pitch adapters) or those coming from the PCB (control, power, monitoring). In all but a small percentage of modules, the damaged bond wires were able to be repaired, thanks to the CERN wire bonding lab, which had excellent technical experience with very difficult bonding situations. It was clear from the way the wires were damaged that the operators had difficulty avoiding touching the area during the mounting of the module. The location of the mounting screws was very close to the wire bonds in many cases. Given that the mounting of the modules in other facilities resulted in much less damage, it was made clear that either a lack of training, a less safe method of mounting, or the need for performing the job very quickly led to this massive damage. If the CERN wire bond lab had not been nearby (it was downstairs in the same building as the module mounting) and available for these often-difficult repairs, there would not have been enough spare modules to complete the endcap installation on time. However, another lesson learned from this was that the encapsulation of those wires that could be safely encapsulated could have prevented much (but not all) of the installation damage described above.

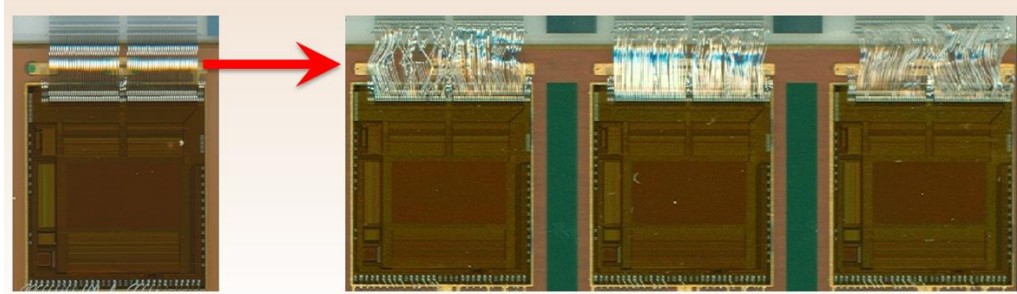

**Figure 13. Left**: Normal row of bond wires. **Right**: Damaged wires on three chips during installation.

### 3.4. Lesson 4: Failure of Conductive Epoxy Backplane Bias Connections

The original plan for making the modules had spots of conductive epoxy placed on gold pads on the polyimide HV bias circuits so the backplanes of the sensor could be biased at high voltage. When the sensor was being picked and placed and glued to the frames (which already had the HV circuits attached), the backplane connection would therefore be made at the same time. However, after time and thermal cycling tests, it was found that an increased resistance was observed in this backplane bias contact and further studies showed that the problem was likely to be silver particle (the conductive agent in these epoxies) migration or oxidation. This was causing the electrical contact in the conductive epoxy to degrade and in some cases to even go to very high resistances or open circuits. As

there was no solution found to avoid this problem, it was decided to wire bond from the aluminium backplane of the sensors to a bondable metallized pad on the HV bias circuits. This meant that every module needed this additional wire bonding step performed after the module was assembled by the gantry robot. This was not a very big problem since every module needed wire bonding at that stage anyways. However, special support jigs were needed to perform this extra wire bonding since this was needed on the back side of the module and not the front as for all the other wire bonds. The wire bonding provided an even better low-resistance contact, which did not degrade with time or thermal cycling.

The problem of loss of conductivity in conductive epoxy connections has not been solved for silver-loaded epoxies as far as I am aware, so this lesson remains quite current for the present and future detectors. One should be very careful to conduct reliability testing using (at a minimum) the worst-case extreme conditions for the environment that the device will experience. This is especially relevant for conductive epoxies but applies to all parts of a detector that have not been reliability tested previously in this way.

### 3.5. Lesson 5: ESD Damage to Modules

Luckily, this problem did not lead to the loss of very many modules as it happened during the very start of production and was caught fairly quickly. Despite the fact that a number of precautions were taken at all the module assembly centers to prevent ESD damage to the modules, one unexpected example of damage occurred during the module assembly process. The damage was discovered as a very high noise in a tight cluster of channels located in the center of the module. As several modules had a very similar problem, the assembly center correctly stopped production and tried to understand the problem. As this problem was presented to the working group on module assembly, it was pointed out that the location and size of the noisy channels looked to be where the vacuum pick-up tool touched the sensors during the assembly. It was then found that this was indeed the cause of the damage and the probable cause of the electrostatic build-up was when the operator cleaned the tool after the previous module assembly and used a standard synthetic or cotton cloth with some ethanol. As the tool surface was made of Teflon, this cleaning likely charged the tool surface. The fix was easy; the use of a proper ESD-safe cleaning cloth and an ESD-safe cleaning fluid removed the problem. It was also possible to replace the pick-up tool with one that had an ESD-safe surface (which is what some of the other assembly centers used). The lesson was that it is very easy to introduce ESD damage if one is not careful at all steps in the assembly procedure. Any object that can touch or get close to the sensors, the bond wires, or the front-end electronics must not get statically charged. This was known to all the personnel involved but somehow this ESD problem was missed.

### 3.6. Lesson 6: The Importance of and the Sometimes Heavy Load of Logistics

Because of the decision of the collaboration to decentralize the construction of the detector, this automatically implied a much greater need for logistics since there would be components, modules, and other detector parts going to and from many sites over the world. Focusing just on module production, the approximate numbers of industrial component producers were: two for sensors, two for pitch adapters, three for ASIC and custom chip production, two for front-end hybrids, two for carbon-fiber frames, and two for HV circuits. For assembly work, the approximate numbers were: 4 sites for sensor testing, 2 sites for chip testing, 2 for hybrid+pitch adapter assembly (and wire bonding), 7 sites for module assembly, 14 sites for module wire bonding, 8 sites for module testing, and 5 sites for module integration on the support structure. The sites for all these activities were mostly in Europe (spread over at least 10 countries), but many were also in the US and a few in Asia. This clearly required many resources to handle just the transport between sites. However, the logistics tasks were not limited to transport, and they also included (and this is not exhaustive): storage facilities, packing, unpacking, sorting, identification, tracking, tooling inventory, consumables, maintenance of equipment and infrastructure,

cleaning, working space, inventory, deciding resource allocation, and deciding component and module distribution flow. Some of these items are particular to a site but some of them relate to the whole project. Clearly the logistics tasks were distributed to the local site managers for the former and there was one person with assistance from a small group dedicated to the project-related logistics. Many of these logistics tasks are best handled by means of a database, so a custom production database was designed and operated successfully for the project. The database itself was a sizeable task but it was sometimes found not to be suitable for all cataloging, tracking, and archival needs. A number of additional local databases were created to better handle specific subtasks. One example was the wire bonding database, which needed to record a very large number of test results (mostly from wire bond pull tests), which did not have general relevance to the project but in case of problems could have been very valuable in solving wire-bonding issues. This was another large task that was not foreseen in the early stages of the project.

From my perspective on the hybrid assembly and wire bonding in particular, the expectation for the resources needed for logistics-related work was greatly underestimated prior to the start of production. In other words, we realized as we started the job that we would need more people, more training, more equipment, more time, and more financial support in order to complete the project successfully. I think this was true more generally in many other areas throughout the tracker project. Thankfully, it was possible to get the additional support needed despite the tight schedule and thus the project was successfully completed.

## 4. Conclusions

The many lessons learned during the CMS tracker module production, including those learned from unexpected problems, were in many cases potentially avoidable or at least could have been reduced in severity had a more stringent and complete quality assurance (QA) plan been applied. At the time, QA was not a commonly used tool in high-energy physics, although it was well-known in industry and in high-reliability projects such as in space programs. So, another general lesson learned was that QA should be better understood in these large-scale projects and applied where it is needed. As industry is well-acquainted with QA (and one should work only with ISO-9000/9001-certified companies if one wants to better avoid QA surprises), the client (us) should also be familiar with QA as it is applied in industry and should know that one must clearly require a QA plan from industrial partners and request from them further QA measures if it is deemed necessary. In addition, the overall QA plan should include the collaborating institutes that are participating in building the detector components. Although there will always be "surprises", a good QA plan includes providing for contingencies in resources and time for the unexpected and should also require more checks and problem avoidance measures in the design and prototyping stages such that fewer surprises occur during production. In my view, this has been the case for the more recent large HEP projects; in particular, the current LHC upgrade projects have learned much from these past lessons. Some of the very unpredictable problems may not be avoided by a good QA plan and, for this, the need for significant contingencies in both resources and time is another key lesson learned from this project.

**Funding:** This research received no external funding.

**Acknowledgments:** I would like to acknowledge the hard work and excellent efforts made by the CMS tracker module production groups including the many dedicated people that designed, produced, and tested the many components of the modules. I would also like to thank K. Klein and M. Mannelli, my CMS colleagues who worked with me on this project, for fact checking, corrections, and helpful comments.

**Conflicts of Interest:** The author declares no conflict of interest.

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
