# Peer review of "Lessons Learned from the Module Production for the First CMS Silicon Tracker"

_instruments, doi:10.3390/instruments6040073_

Round 1

Reviewer 1 Report

The article report the experience gained by author and by the collaboration during the production (construction and assembly) of the first silicon tracker of CMS, with a focus on the main problems that required special, and not foreseen initially, measurements.

The article reports, in a clear and plain language, a selection of issues the collaboration faced during the production of the modules of the CMS tracker. The choice for sure, and obviously, suffer from the author bias, but that was declared honestly since the title (“Some lessons”), and the article never claims to be a comprehensive review.

The text is clear and and readability is very good. English is not my mother tongue, so I may have overlooked something, but I really I have almost nothing to comment:

- lines 63-64: I feel it should be figure 4 and figure 5 (and not 3 and 4)) 

- line 323: missing space between “2” and “mm”

- line 451: missing space between “200” and “Hz”

- line 476: “(typically 100 ppm/°K compared to 3 ppm/°K for silicon)”. Correct to either “ppm/K” or “ppm/°C”

(and, I realised here, I have the impression that in all the occurrences, there is not the due space between the numeric value and the °C symbol )

- lines 508-509:  the sentence “The Mounting…” seem to me repeating what already said, with other words, at lines 503-505 (“One endcap …”)

The figure are, generally, of good quality, but I’ve some concerns about some of the superimposed text in Figures 3, 8, and 12. The color choice is not always providing a good contrast. Viewing them to grayscale (miming the BW printing), in some cases the visibility improves, but in general it is getting a little worse. 

In two cases (the rightmost image, in both Figs. 8 and 12) the text is even a bit blurred.

In all cases, the text superimposed to the pictures is readable, even if sometime with some extra effort. For this reason I suggest the author, IF the original pictures are available, to improve the mentioned cases. If, due the long time since the reported events, they are not, I’m fine in keeping what already presented.

The bibliography is minimal, nevertheless, due to the topic of the article I found it OK, and I’m fine with the author choice.

In conclusion, I found the article extremely interesting, and I believe it is important that such complex of small but fundamental details are documented and public available, avoiding the lesson should be learned again.

I’m completely in favour of the publication as soon as the minor flaws are fixed

Author Response

Hello and thank you for your helpful review.

As the editor may have explained, you had received an older version of my article to review. The newer version has some significant changes, many corrections to text but also some removal of parts of sections and addition of a new section (a new “lesson”). These were required changes since I forgot to go through the editing and reviewing process of the CMS collaboration. I am sorry that you may need to re-review my article because of this. Most of the text is the same as the version you read but the largest changes were that lesson #1 had a paragraph removed and one new section was added (this meant one new "lesson" was added).

However, I can say that many of your requested corrections had been already done in the newer version and I have made most of your requested corrections in the very latest version. Here are the point-by-point responses (note that line numbers are likely now different):

lines 63-64: this is corrected

line 323: this is corrected

line 451: this is corrected

line 476: this is corrected

space before the symbol degree C has been inserted

lines 508-509: this section has been removed so that text no longer exists

text in figures 3, 8 and 12: Most of these have been redone to make the text font larger and clearer. There are some cases of text in a photo that cannot be modified as I don't have a version of the photo without the text. However, I believe all cases can be magnified (in the case of on-line reading where one can magnify the view of the article) so that the text can be read, if needed.

Best regards,

Alan Honma

Reviewer 2 Report

Well written publication, can be published in the present form. Please improve

the letter character size and colour on figures, eg fig.3, fig.8. The reader cannot read easily what it is highlighted.

Author Response

Hello and thank you for your helpful review.

As the editor may have explained, you had received an older version of my article to review. The newer version has some significant changes, many corrections to text but also some removal of parts of sections and addition of a new section (a new “lesson”). These were required changes since I forgot to go through the editing and reviewing process of the CMS collaboration. I am sorry that you may need to re-review my article because of this. Most of the text is the same as the version you read but the largest changes were that lesson #1 had a paragraph removed and one new section was added (this meant one new "lesson" was added).

However, I can say that your requested corrections had been already done in the newer version. Here is the point-by-point response:

All figures that had too small or difficult to read text in the photos or drawings were revised in the newer version of my article. There was one case where the text was not changeable since I did not have the original photo without the text. However, I think that because one can easily magnify the view of the article (when viewing the pdf version), one should be able to read all cases where the text is a bit small.

Best regards,

Alan Honma

Reviewer 3 Report

Dear Author,

  thanks for summarising these "stories". HEP experiments are actually to be managed with schemes borrowed from industry and your paper is teaching the community a lot.

Author Response

Hello and thank you for your review.

As the editor may have explained, you had received an older version of my article to review. The newer version has some significant changes, many corrections to text and figures but also some revision of sections and addition of a new section. These were required changes since I forgot to go through the editing and reviewing process of the CMS collaboration. I am sorry that you may need to re-review my article because of this. Most of the text is the same as the version you read but the largest changes were that lesson #1 had a paragraph removed and one new section was added (this meant one new "lesson" was added).

Best regards,

Alan Honma

Round 2

Reviewer 1 Report

The updated and corrected of the article is of very good quality (no surprise since the first version already was). I recognise the effort of the author to improve the pictures in all cases this was possible.  

As written in revision_1  the article is well written and  very interesting. I liked the "extra" lesson that was added. Beside this extra paragraph, the new version fixed the few flaws and typos of the first version. Just one got missed: a space between "°" and "C" at line 463

I'm completely in favour of the publication in the present form.

Reviewer 2 Report

Revision completed. Well written publication.

Reviewer 3 Report

Thanks for making the final draft available.

I stay with what I stated beforehand.

Well done!